# SB365, *Pulsatilla* Saponin D Induces Caspase-Independent Cell Death and Augments the Anticancer Effect of Temozolomide in Glioblastoma Multiforme Cells

**DOI:** 10.3390/molecules24183230

**Published:** 2019-09-05

**Authors:** Jun-Man Hong, Jin-Hee Kim, Hyemin Kim, Wang Jae Lee, Young-il Hwang

**Affiliations:** 1Department of Anatomy and Cell Biology, Seoul National University College of Medicine, Seoul 03080, Korea (J.-M.H.) (W.J.L.); 2Department of Biomedical Laboratory Science, Cheongju University, Cheongju 28503, Korea; 3Research Institute for Future Medicine, Samsung Medical Center, Seoul 06351, Korea

**Keywords:** *Pulsatilla* saponin D, SB365, glioblastoma multiforme, temozolomide, autophagic flux inhibition, lysosomal membrane permeabilization, mitochondrial membrane potential

## Abstract

SB365, a saponin D extracted from the roots of *Pulsatilla koreana*, has been reported to show cytotoxicity in several cancer cell lines. We investigated the effects of SB365 on U87-MG and T98G glioblastoma multiforme (GBM) cells, and its efficacy in combination with temozolomide for treating GBM. SB365 exerted a cytotoxic effect on GBM cells not by inducing apoptosis, as in other cancer cell lines, but by triggering caspase-independent cell death. Inhibition of autophagic flux and neutralization of the lysosomal pH occurred rapidly after application of SB365, followed by deterioration of mitochondrial membrane potential. A cathepsin B inhibitor and *N*-acetyl cysteine, an antioxidant, partially recovered cell death induced by SB365. SB365 in combination with temozolomide exerted an additive cytotoxic effect in vitro and in vivo. In conclusion, SB365 inhibits autophagic flux and induces caspase-independent cell death in GBM cells in a manner involving cathepsin B and mainly reactive oxygen species, and its use in combination with temozolomide shows promise for the treatment of GBM.

## 1. Introduction

Glioblastoma multiforme (GBM) is the most frequent and most malignant brain tumor, with a mean survival of GBM patients of less than 2 years [1]. Although several therapeutic modalities including immunotherapies are under development [2], the standard therapy for newly diagnosed GBM is surgical resection within a maximum range followed by concomitant chemotherapy and radiotherapy [2,3]. For chemotherapy, temozolomide (TMZ) is the drug of choice [4]. TMZ is an oral alkylating agent that induces DNA methylation at the O^6^ position of guanine. The resultant O^6^-methylguanine is abnormally paired with thymine, leading to cleavage of DNA strands by the mismatch-repair system, which triggers apoptosis [5]. TMZ is suitable for treating GBM because it can pass the blood–brain barrier [6]. However, resistance to TMZ can be induced in GBM cells by expression of *p*53, *p*21, or O^6^-methylguanine-DNA methyltransferase (MGMT) [7]. Furthermore, TMZ has side effects such as genotoxicity, fetal toxicity, and lymphocytopenia of T cells and NK cells [8].

Combinations of drugs are typically used to reduce the likelihood of toxicity and side effects [9]. In patients with GBM, combinations of TMZ with inhibitors of autophagic flux (e.g., hydroxychloroquine) have been developed, on the basis that blocking autophagy should enhance the effects of TMZ because autophagy protects against the toxicity of radiotherapy and TMZ [10]. However, such combinations can cause side effects such as anemia, maculopapular rash, hemolysis, and decreased platelet and immune cell counts [10].

SB365 is a saponin D, hederagenin 3-*O*-α-l-rhamnopyranosyl(1→2)-(β-d-glucopyranosyl(1→4))-α-l-arabinopyranoside, which is extracted from the roots of *Pulsatilla koreana* [11]. Among eight lupane- and nine oleanane-type saponins extracted from *P. koreana*, SB365 showed the greatest antitumor activity in vitro against A-549 (lung cancer), SK-OV-3 (ovarian cancer), SK-MEL-2 (melanoma), and HCT-15 (colon cancer) cells. Indeed, its effect was superior to those of Taxol and doxorubicin [12]. In immunocompromised mice, SB365 suppressed the proliferation of human Huh-7 (liver cancer), MKN-45 (gastric cancer), PANC-1 (pancreatic cancer), and HT-29 (colon cancer) cells, without weight loss or toxicity to normal tissue [13,14,15,16]. In a clinical trial involving patients with stage IV pancreatic cancer, SB365 increased the survival rate without inducing side effects [17].

SB365 is reported to induce apoptosis of cancer cells in vitro [13,14,15,16,18] and to inhibit the autophagic flux in HeLa (cervical cancer), K562 (leukemia), B16-F10 (melanoma), A549 (lung cancer), and MCF-7 (breast cancer) cells. Moreover, SB365 additively enhanced the anticancer activity of the chemotherapeutic agents 5-fluorouracil, camptothecin, and etoposide in HeLa cells in vitro [19].

The effects of SB365 on GBM cells have, to our knowledge, not yet been investigated. Furthermore, if it inhibits autophagic flux in GBM cells, SB365 in combination with TMZ could be used for the treatment of GBM, replacing chloroquine or hydroxychloroquine.

The aim of this study was to investigate the effects of SB365 alone and in combination with TMZ on GBM cells in vitro and in vivo. To this end, we selected two GBM cell lines, U87-MG and T98G, among dozens of them. These are of human grade IV glioma cells [20]. We selected them because they are the most extensively employed ones in related studies [21], and especially they possess opposite characteristics to the susceptibility to TMZ. U87-MG cells are susceptible to TMZ, while T98G cells are not. T98G cells express O^6^-methylguanine-DNA methyltransferase (MGMT), which removes the methyl group at the O^6^ position of guanine added by TMZ [22], rendering them resistant to this drug. The survival duration of patients with MGMT-expressing GBM is approximately two years less than that of patients with non-functional methylated MGMT genes [23].

## 2. Results

### 2.1. SB365 Inhibited the Proliferation of GBM Cells In Vitro

The proliferation of U87-MG cells treated with SB365 was assayed after 24, 48, and 72 h (Figure 1). At 24 h, cell proliferation was comparable to that of the control group (Figure 1A), irrespective of SB365 concentration. However, after 48 h, 20 μM SB365 reduced cell proliferation by ~30% compared to the control (Figure 1B). After 72 h, 2.5 and 20 µM SB365 reduced cell proliferation by 25% and 80%, respectively, compared to the control (*p* < 0.001) (Figure 1C). Similar results were obtained using TMZ-resistant T98G cells (Appendix A). Calculated half maximal inhibitory concentration (IC50) for 72 h treatment was 8.9 μM.

Moreover, after 24 h, flow cytometry showed that SB365 did not significantly increase the frequency of annexin V-positive cells (Figure 1E and Appendix A). After 48 h, 20 µM SB365 resulted in a significant increase in the frequency of annexin V-positive cells (Appendix A). After 72 h, the frequency of annexin V-positive cells increased by 2.5–20 µM SB365 in a dose-dependent manner (Figure 1D,E). Similar results were obtained using TMZ-resistant T98G cells (Appendix A).

### 2.2. SB365 Induced the Death of GBM Cells in a Caspase-Independent Manner

The cytotoxic effect of SB365 in cancer cells is mediated by apoptosis [13,14,15,16,18]. Since FACS showed the presence of few cells in the early stage of the apoptotic process, which are 7-AAD-negative and annexin V-positive [24], we furthered explored SB365-induced apoptosis of U87-MG cells.

The level of cleaved caspase-3, the final caspase of the intrinsic and extrinsic apoptosis pathways [25], in cells treated with 10 μM SB365 for 72 h was evaluated by western blotting (Figure 2A,B). SB365 triggered cleavage of caspase-3 in HT-29 and Huh-7 cells, as reported previously [13,14], but not in U87-MG cells. When the cells were stained with DAPI, SB365-treated HT-29 and Huh-7 cells showed nuclear blebbing and/or fragmentation with a frequency of 1–4 nuclei per a high-power field. However, SB365-treated U87-MG cells showed round or oval nuclei without blebbing and fragmentation (Figure 2C). Thus, SB365 induced caspase-independent cell death (CICD) rather than caspase-dependent apoptosis in U87-MG cells. Similar results were obtained using T98G cells (Appendix A).

### 2.3. SB365 Induced Autophagic Flux Inhibition in GBM Cells

SB365 reportedly inhibits autophagic flux in HeLa, K562, A549, and MCF-7 cells [19]. Given that autophagy protects against cell damage [26], its inhibition could be involved in SB365-induced death in GBM cells. Thus, we evaluated whether SB365 inhibited autophagic flux in U87-MG cells.

The cells were treated with 10 μM SB365, and the expression of microtubule-associated protein light chain 3 (LC3)-I, II, and p62 was evaluated by western blotting within 24 h. When autophagy is induced, LC3-I is converted to LC3-II in combination with phosphatidylethanolamine in the cytosol to produce autophagosomes, and *p*62 binds to ubiquitinated proteins and pulls them into autophagosomes to be decomposed due to subsequent autophagic flux [27]. When the autophagic flux is inhibited, LC3-II and p62 accumulate in the cell [28]. Thus, the LC3-II/I ratio and *p*62 were regarded as indicators of autophagic flux inhibition.

The *p*62 level and LC3-II/I ratio (Figure 3A,B) increased in a time-dependent manner, indicating that SB365 inhibits autophagic flux. The *p*62 level and LC3-II/I ratio in U87-MG and T98G cells remained high until 72 h (Appendix A), but the expression of beclin-1 did not change significantly (Figure 3 and Appendix A).

### 2.4. Inhibition of Autophagic Flux by SB365 is Linked to Lysosomal Neutralization and Reduction of MMP

Since inhibition of autophagic flux is associated with lysosomal dysfunction such as neutralization and permeabilization [29], we performed a lysosomal stability test. Cells were stained with acridine orange and analyzed by flow cytometry. The frequency of cells emitting red fluorescence decreased by 65% at 6 h after SB365 treatment compared to the control and decreased steadily thereafter (*p* = 0.05) (Figure 4A,C).

Next, we measured alterations in mitochondrial membrane potential (MMP), which typically occur after lysosomal dysfunction [30]. Cells were treated with 10 μM SB365 as above, stained with JC-1 for 20 min, and analyzed by flow cytometry. The frequencies of cells with altered MMP were 5.8% and 8.6% higher at 36 and 48 h after SB365 treatment, respectively, compared to the control (*p* = 0.01) (Figure 4B,C).

### 2.5. Cathepsin B and Reactive Oxygen Species Contribute to SB365-Induced Cell Death

Since lysosomal membrane permeabilization (LMP) is a frequent cause of lysosomal dysfunction, and leads to leakage of cathepsin B and/or cathepsin D from the lysosome into the cytoplasm, resulting in cell death [31,32,33], we determined whether SB365-induced cell death was due to leakage of cathepsins. To this end, cell proliferation was evaluated 72 h after SB365 treatment in the presence or absence of cathepsin inhibitors. A cathepsin B inhibitor II at 5 μM recovered the cell proliferation inhibited by SB365 by ≥40% (*p* = 0.05) (Figure 5A) and reduced the frequency of cells with altered MMP (Figure 5B). However, a cathepsin D inhibitor (pepstatin A) exerted no such effects (data not shown).

Next, we evaluated whether reactive oxygen species (ROS) were related to SB365-induced cell death, because autophagic flux inhibition [34,35] and MMP deterioration [36] increase intracellular ROS levels, leading to cell death. Cells were treated with the indicated concentrations of the antioxidant *N-*acetyl cysteine (NAC) 1 h after SB365 exposure. After 72 h, NAC recovered the suppression of proliferation caused by SB365 (by ~30% at 0.625 mM and 50% at 2.5 mM) (Figure 5C). However, NAC at 5 mM did not recover the inhibition of cell proliferation. NAC exerted a similar effect in T98G cells, albeit to a lesser degree (Appendix A). Considering that MMP deterioration started late during the experiment time (Figure 4B,C) and that NAC could decompose in culture media, we performed the same experiment with 2.5 mM NAC, which at this time was added 24 and 48 h after SB365 treatment, instead of 1 h after (Figure 5D). As a result, NAC recovered the cytotoxicity by SB365 up to over 70% when added at 48 h.

### 2.6. SB365 and TMZ Additively Inhibited the Proliferation of GBM Cells In Vitro

Since SB365 inhibited autophagic flux in GBM cells, we evaluated its influence on the anticancer activity of TMZ, like other autophagic flux inhibitors such as hydroxychloroquine [10].

U87MG cells were treated with TMZ in the presence or absence of 10 μM SB365 for 72 h, and their proliferation was determined by CCK-8 assay. TMZ alone at 25 and 50 μM inhibited cell proliferation by 37% and 46%, respectively, compared to the control (*p* < 0.001) (Figure 6A). Lower concentrations of TMZ (6.25 and 12.5 μM) also inhibited cell proliferation, albeit not significantly. The combination of TMZ (6.25, 12.5, 25, and 50 µM) and SB365 inhibited cell proliferation by 46%, 48%, 56%, and 63%, respectively (*p* = 0.016) (Figure 6B). Similar results were obtained using T98G cells (Appendix A). At low TMZ concentrations, the combination exerted an additive effect on cell proliferation. That is, the combination of 10 µM SB365 with 6.25 and 12.5 µM TMZ increased the inhibition of cell proliferation from 6% to 46%, and from 10% to 48%, respectively (Appendix A).

### 2.7. SB365 Inhibited Tumor Growth in the Mouse U87-MG Xenograft Model

Based on the above in vitro results, the effects of SB365 and/or TMZ on tumor growth in vivo were investigated. U87-MG cells were inoculated into both flanks of nude mice. When the tumor volume reached 100–200 mm^3^, SB365 (5 mg/kg/every other day, intratumoral) and/or TMZ (2.5 mg/kg/day, intraperitoneal) were administered until day 22. The doses were determined based on previous reports and the results of a pilot study (data not shown). No marked change in body weight was detected (Figure 7A).

Tumor growth was significantly inhibited by injection of SB365 or TMZ only compared to the control (*p* = 0.011) (Figure 7B). In addition, the combination of SB365 and TMZ resulted in significantly greater inhibition of tumor growth compared to TMZ or SB365 only (*p* = 0.046) (Figure 7B). The tumor weights at the end of the experiment were in agreement with these results (Figure 7C,D).

## 3. Discussion

In this experiment, SB365 exerted a cytotoxic effect on these cells in a dose-dependent manner. However, this effect was mediated by induction of, not apoptosis, as in other cancer cells, but CICD. The cytotoxic impact of SB365 proceeded as follows: neutralization of the lysosomal pH and inhibition of autophagic flux occurred rapidly, followed by alteration of MMP, and finally, cell death. SB365-induced cell death was partially recovered by treatment with a cathepsin B inhibitor and NAC. Moreover, the combination of SB365 and TMZ exerted an additive effect both in vitro and in vivo.

SB365 is administered intratumorally via direct percutaneous injection to patients with pancreatic cancer [17]. To mimic this, we injected the agent directly into the tumor mass in mice, rather than administering intraperitoneally or orally, as in prior studies [13,14,15,16].

The dose-dependency of the cytotoxic effect (Figure 1) of SB365 is in agreement with prior findings in liver, lung, colon, and pancreatic cancer cells [13,14,15,16,18]. SB365 induced caspase-3 cleavage and nuclear fragmentation in colon cancer and hepatocarcinoma, but not in GBM cells (Figure 2). Activation of caspase-3 is a converging step of both the intrinsic and extrinsic pathways of caspase-dependent apoptosis [37]. In addition, SB365 did not affect Bcl-2 and Bax expression in U87-MG cells (data not shown) the expression of which decreases and increases, respectively, during initiation of apoptosis [38]. Thus, we assumed that SB365 induced CICD in GBM cells.

To evaluate the mechanism underlying SB365-induced death in GBM cells, we explored its effect on autophagic flux, because CICD in GBM cells by chloroquine [33] and thymoquinone [31] is associated with inhibition of autophagic flux, and SB365 inhibits autophagic flux in other cancer cell lines [19]. The levels of LC3-II and *p*62 increased at 6 h after SB365 treatment (Figure 3A,B), and remained high up to 72 h (Appendix A
Appendix A), which implies that the SB365-induced death of GBM cells may be associated with inhibition of autophagic flux.

SB365 induces autophagy in HeLa cells by increasing ERK phosphorylation and decreasing mTOR activation, though it inhibits subsequent autophagic flux [19]. In hepatocarcinoma [13] and gastric cancer [14] cells, SB365 suppressed the PI3K/Akt/mTOR pathway, which negatively regulates autophagy [39]. However, in U87-MG cells, the *p*-Akt and *p*-mTOR levels were unchanged after 24 h of treatment with SB365 (data not shown). Furthermore, the cytotoxic effect of SB365 on U87-MG cells was augmented by pretreatment with a non-toxic concentration of the autophagy inducer rapamycin [40] (Appendix A). These results imply that the accumulation of autophagosomes due to inhibition of the autophagic flux caused cell death. Critically, SB365 did not increase the expression of beclin-1 (Figure 3 and Appendix A), which is associated with autophagy induction [41]. Therefore, SB365 does not induce autophagy, but inhibits autophagic flux, in U87-MG cells.

Inhibition of autophagic flux can result from lysosomal neutralization [29]. SB365 treatment resulted in simultaneous inhibition of autophagic flux (Figure 3) and lysosomal neutralization (Figure 4A,C). Thus, the SB365-induced inhibition of autophagic flux may be mediated by lysosomal deterioration. Indeed, saponins, in particular oleanane-type saponins such as SB365 [12], reportedly permeabilize the cell membrane [42] and the lysosomal membrane [43]. In addition, a cathepsin B inhibitor partially restored the SB365-induced reduction in cell proliferation (Figure 5A), suggesting that cathepsin B was released from lysosomes and that SB365 induced permeabilization of the lysosomal membrane.

In our results, a cathepsin B inhibitor restored cell death but a cathepsin D inhibitor did not (data not shown). Given that the molecular weights of cathepsins B and D are similar [44], and thus the two molecules would have been released simultaneously, the contradictive effect of each inhibitor would be somewhat unexpected. However, the same results have been reported in paclitaxel-, epothilone B-, and discodermolide-treated human non-small cell lung cancer cells [45] and supraoptimally activated T cells [46]. Possibly, only cathepsin B had been released [44]. Alternatively, these results suggest the varying role of cathepsins depending on the type of cells [30]. The exact mechanisms remain to be determined.

The frequency of the cells with MMP deterioration was only 5.8% at 36 h and 8.6% at 48 h after SB365 treatment (Figure 4B,C). These are low values considering that MMP deterioration directly led to the SB365-induced cell death. Indeed, the phenomena caused by various factors secreted from the mitochondria when MMP deterioration occurs, such as the activation of caspase-3/9 leading to apoptosis by cytochrome c, degrading DNA by endonuclease G, and chromatin condensation by AIF [47] were not observed in this experiment. Another substance that is released from deteriorated mitochondria is ROS. Autophagic flux inhibition, which was induced by SB365 in GBM cells in this experiment, leads to the accumulation of ROS [34,35]. Excess ROS accelerate lysosomal permeabilization, and leaked lysosomal proteases deteriorate MMP, resulting in increased cytoplasmic ROS leakage, creating a vicious cycle [48]. Thus, ROS could be a factor for the SB365-induced cytotoxicity. Substantial to this assumption, 2.5 mM NAC recovered the cytotoxicity by over 50% when added 1 h after SB365 treatment (Figure 5C). Furthermore, when NAC was added 48 h after SB365 treatment, the recovery rate was over 70% (Figure 5D). These results imply that ROS was the main factor leading to cell death by SB365, and ROS presumably began to accumulate to cause cell death 24 h after SB365 treatment in parallel with MMP deterioration.

Meanwhile, 5 mM NAC failed to recover cell proliferation. This may be because of excessive eradication of ROS by the antioxidant, which performs physiological functions in cell proliferation [49]. Substantial to this assumption, 10 mM NAC augmented the cytotoxic effect of SB365 (data not shown). Additionally, even the low concentrations of NAC (0.623–2.5 mM) augmented the effect of SB365 when it was treated before SB365 (data not shown).

Attempts have been made to improve the efficacy of TMZ against GBM by combining it with other drugs. TMZ is typically combined with autophagic flux inhibitors such as chloroquine, hydroxychloroquine, or bafilomycin A1, with which it reportedly exerts synergistic effects [10]. Since SB365 inhibited autophagic flux in GBM cells, we evaluated the efficacy of the combination of SB365 and TMZ. The combination of SB365 and TMZ increased the frequency of cell death in vitro (Figure 6) and inhibited tumor growth in vivo (Figure 7). Thus, SB365 could be used in combination with TMZ in place of chloroquine, hydroxychloroquine, or bafilomycin A1, which synergistically inhibit tumor growth but have several side effects [10,50]. One concern is that SB365 exerts hemolytic activity on red blood cells of the sheep [42] and the rabbit [51], which was considered as a major drawback for its clinical development [42].

SB365 alone induced death in TMZ-resistant T98G cells (Appendix A) as effectively as in TMZ-sensitive U87-MG cells. Furthermore, in T98G cells, the combination of SB365 and TMZ additively increased cell death (Appendix A). Unfortunately, we did not determine whether SB365 downregulated the expression of MGMT genes.

In conclusion, SB365 inhibited autophagic flux, and induced CICD in GBM cells in a manner mediated by cathepsin B and mainly by ROS very likely due to autophagic flux inhibition and MMP deterioration. Moreover, SB365 and TMZ exerted an additive cytotoxic effect in vivo and in vitro. Thus, SB365 could be used in combination with TMZ for the treatment of TMZ-resistant GBM.

## 4. Materials and Methods

### 4.1. Chemicals

SB365 was supplied by SB Pharmaceutical Co. Ltd. (Gongju, Republic of Korea (ROK)). TMZ (T2577) was purchased from Sigma-Aldrich (St. Louis, MO, USA). SB365 and TMZ stock solutions (100 mM) were prepared in dimethyl sulfoxide (DMSO). The final DMSO concentration in culture media was ≤0.4%, which did not exert a toxic effect on GBM cells (data not shown). Stock solutions of cathepsin B inhibitor II (219385; Calbiochem, San Diego, CA, USA), pepstatin A (cathepsin D inhibitor, P5318; Sigma-Aldrich, and *N*-acetyl cysteine (NAC) (A7250; Sigma-Aldrich, Saint Louis, MO, USA) were prepared and stored at −80 °C until use.

### 4.2. Cell Lines and Culture Conditions

TMZ-susceptible U87-MG and TMZ-resistant T98G human GBM cells, as well as HT-29 and Huh-7 cells (Korean Cell Line Bank, Seoul, ROK) were used in this study. The cells were cultured in minimum essential Eagle’s medium (EMEM) supplemented with 10% fetal bovine serum, 1% penicillin/streptomycin, and 1% non-essential amino acids (Welgene, Daegu, ROK) at 37 °C in a 5% CO_2_ atmosphere in a humidified chamber.

### 4.3. Cell Counting Kit-8 Assay

The cytotoxicity of SB365 and TMZ was assessed using a Cell Counting Kit-8 (CCK-8; EZ-3000; Dojindo, Kumamoto, Japan) following the manufacturer’s instructions. Briefly, U87-MG cells (5 × 10^3^/well) or T98G cells (2 × 10^3^/well) were cultured in quadruplicate in 96-well plates overnight and treated with SB365 and/or TMZ at predefined concentrations. The culture medium was discarded, and 100 μL of CCK-8 working solution (10% (*v*/*v*) CCK-8 stock solution in phosphate-buffered saline (PBS)) were added. The cells were incubated at 37 °C for 1–3 h, and the absorbance at 450 nm was measured using a SpectraMax Plus 384 spectrophotometer (Molecular Devices, Sunnyvale, CA, USA).

IC50 value was obtained, based on the CCK-8 results, by the Quest Graph™ IC50 Calculator, a four parameter logistic regression model [52], with the minimum response value fixed to zero computationally.

### 4.4. Apoptosis Assay

U87-MG (7.5 × 10^4^/well) and T98G (3 × 10^4^/well) cells were seeded in a six-well plate and cultured overnight at 37 °C in a CO_2_ incubator. The cells were treated with SB365 and/or TMZ and collected in fluorescence-activated cell sorting (FACS) tubes. After washing twice with FACS buffer (0.5% BSA in PBS), the cells were resuspended in 100 μL of FACS buffer, 2 μL of annexin V were added (556419; BD Pharmingen, San Jose, CA, USA) and the plate was shaken for 15 min at room temperature. Next, 1 μL of 7-AAD was added (559925; BD Pharmingen), and the cells were subjected to FACS analysis on a FACSCalibur flow cytometer (BD Biosciences, Heidelberg, Germany).

To evaluate nuclear morphology, cells treated with SB365 for 72 h were harvested and seeded onto poly-l_-_lysine-coated multispot slides. The cells were washed with PBS, fixed in 4% paraformaldehyde for 20 min, and stained with 4,6-diamidino-2-phenylindole (DAPI; F6057, Sigma-Aldrich, Saint Louis, MO, USA).

### 4.5. Western Blotting

Cells were dissociated by pipetting in cold radioimmunoprecipitation assay (RIPA) buffer (50 mM Tris-HCl (pH 7.4), 150 mM NaCl, 1% sodium deoxychloride, 0.1% sodium dodecyl sulfate (SDS), 1% Triton X-100, 2 mM ethylenediaminetetraacetic acid (EDTA), and 1% protease inhibitors), and centrifuged at 18,000× *g* for 10 min at 4 °C. The supernatant was collected, and the protein concentration measured by bicinchoninic acid assay, then 20 or 100 μg (for caspase-3) of protein were mixed with RIPA buffer and 5× SDS loading dye (S2002; Biosesang, Seongnam, ROK) to a final volume of 20 μL. The mixture was boiled at 95 °C for 10 min, loaded onto a sodium dodecyl sulfate polyacrylamide gel, and electrophoresed at 50 V for stacking and 120 V for separation. The samples were subsequently transferred to a nitrocellulose membrane at 400 mA for 1 h at 4 °C and blocked in blocking buffer (5% skim milk, 0.05% Tween 20 in PBS) for 1 h at room temperature. Finally, the samples were incubated with the appropriate primary antibody in blocking buffer overnight at 4 °C, followed by the corresponding secondary antibody for 1 h at room temperature. Protein bands were visualized using an enzyme-linked chemiluminescence detection kit (DG-WF200; DoGEN, Seoul, ROK). The primary antibodies used were as follows: rabbit anti-human LC3B (NB 600-1384; Novus Biologicals, Minneapolis, MN, USA; 1:5000); rabbit anti-human beclin-1 (ab2557; Abcam, Cambridge, MA, USA; 1:5000); mouse anti-human *p*62 (ab56416; Abcam; 1:10,000); rabbit anti-human caspase-3 (9662), *p*-AKT (9271), AKT (9272), *p*-mTOR (2971), and mTOR (2972; Cell Signaling Technology, Inc., Danvers, MA, USA; 1:1000); and mouse anti-human β-actin (3700; Cell Signaling Technology; 1:5000). A goat anti-mouse IgG-horseradish peroxidase (HRP) (SC-2005; Santa Cruz Biotechnology, Santa Cruz, CA, USA; 1:5000) or anti-rabbit IgG-HRP (SC-2030; Santa Cruz Biotechnology; 1:5000) was used as the secondary antibody.

### 4.6. Lysosome Stability Assay

Lysosomal membrane stability was determined by staining SB365-treated cells with 3 μg/mL acridine orange (A8097; Sigma-Aldrich, Saint Louis, MO, USA) for 20 min at 37 °C. This metachromatic dye emits red fluorescence when it is confined in the cytosol where it is present as a monomer. When the dye penetrates into the dysfunctional lysosome, it converts into aggregates due to the acidic environment in the lysosome and emits green fluorescence. The property has been used to measure lysosomal membrane stability [53]. Flow cytometric analysis was performed to determine the red (FL3; 650 nm) and green (FL1; 510–530 nm) fluorescence of cells excited by blue (488 nm) light using a FACSCalibur instrument.

### 4.7. Mitochondrial Membrane Potential Assay

SB365-treated cells were stained with 2.5 μM JC-1 (T3168; Life Technologies, Carlsbad, CA, USA) for 20 min at 37 °C, and analyzed by flow cytometry. JC-1 is a lipophilic and cationic dye. It enters the mitochondria, converts from monomers to aggregates by membrane potential, and accumulates inside the mitochondrion. In FACS analysis, monomers and aggregates emit green and red fluorescence, and indicate lower and higher mitochondrial membrane potential (MMP), respectively [54].

### 4.8. Animal Xenograft Model

Animal experiments were approved by the Institutional Animal Care and Use Committee (SNU-150521-3-2). Seven-week-old male Balb/c-nu mice were purchased from OrientBio (Seongnam, ROK). U87-MG cells were mixed with Matrigel HC (354248; BD Biosciences) at a 50:50 volume ratio, and the mixture was inoculated into both flanks (5 × 10^6^ cells/100 μL/flank) of the mice. When the tumor reached a volume of approximately 100–200 mm^3^, the mice were assigned to control, SB365, TMZ, and SB365 + TMZ treatment groups; the mean mass of each group was similar. Next, the mice underwent intratumoral injection of SB365 (5 mg/kg) every other day and/or intraperitoneal injection of TMZ (2.5 mg/kg) or vehicle (≤3% DMSO) daily. The day of the first injection was regarded as day 0 and the injections were administered until day 21; the mice were sacrificed on day 22. The body weight and tumor volume were measured every other day. Tumor size was measured using calipers and the tumor volume was calculated as volume (V) = length (L) × width (W)^2^ × 0.5.

### 4.9. Statistical Analysis

The Mann–Whitney U-test was used to evaluate statistical significance. Statistical analysis was performed using Statistical Package for the Social Sciences software ver. 12 (SPSS, Inc., Chicago, IL, USA). A value of *p* < 0.05 was taken to indicate statistical significance.

## Figures and Tables

**Figure 1 molecules-24-03230-f001:**
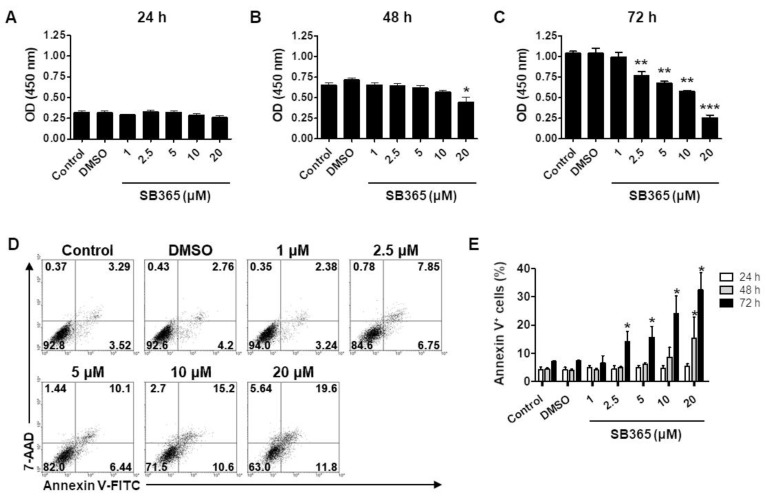
SB365 exerted a cytotoxic effect on U87-MG cells. (**A**–**C**) SB365 inhibited the proliferation of U87-MG cells. The cells in 96-well plates were treated with SB365 at the indicated concentrations for (**A**) 24, (**B**) 48, or (**C**) 72 h in quadruplicate, and subjected to CCK-8 assay. (**D**,**E**) SB365 increased the frequency of the annexin V-positive cells. U87-MG cells in six-well plates were treated as above, stained with annexin V and 7-AAD, and subjected to FACS analysis. (**D**) A representative FACS profile after 72 h and (**E**) the frequency of annexin V-positive cells. Experiments were performed independently in triplicate. * *p* < 0.05, ** *p* < 0.01, and *** *p* < 0.001 vs the control.

**Figure 2 molecules-24-03230-f002:**
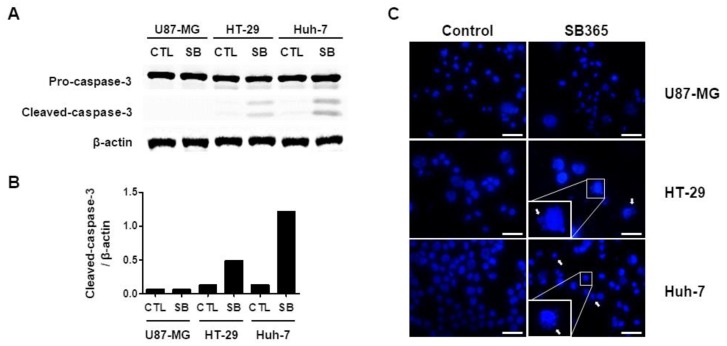
SB365 induced caspase-independent death in U87-MG cells. U87-MG, HT-29 (1 × 10^5^/well), and Huh-7 cells (1 × 10^5^/well) in six-well plates were treated with 10, 5, and 15 μM SB365, respectively. The calculated IC50 values of SB365 on each cell line were 8.9, 5.1, and 13.2 μM, respectively. (**A**) Cell lysates were subjected to western blotting of caspase-3 cleavage, (**B**) followed by densitometry. (**C**) SB365 induced nuclear fragmentation in HT-29 and Huh-7 cells, but not in U87-MG cells. Cells were treated with 10 μM SB365 for 72 h, adhered to an eight-well multispot slide, and stained with DAPI (blue). Arrows indicate fragmented nuclei. Images were acquired using a fluorescence microscope (x 400). The scale bar represents 50 μm. CTL, control group; SB, SB365-treated group.

**Figure 3 molecules-24-03230-f003:**
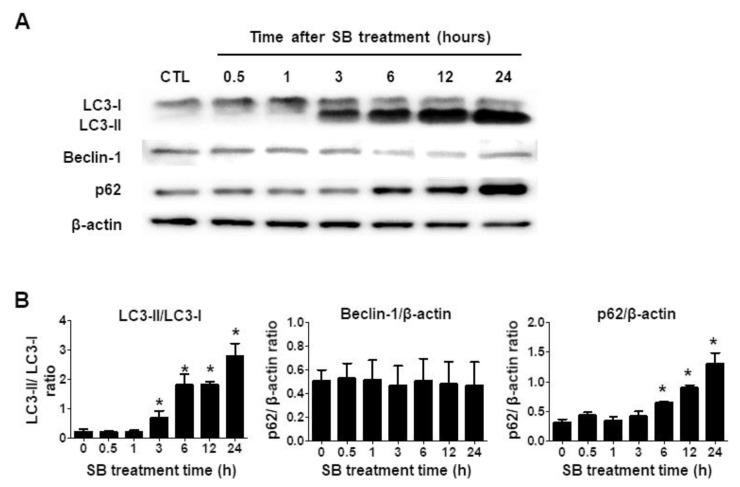
SB365 inhibited autophagic flux in U87-MG cells. Western blot analysis of autophagy-related proteins within 24 h of treatment with SB365. U87-MG cells in a six-well plate were treated with 10 μM SB365 for the indicated times. (**A**) Cell lysates were subjected to western blotting for LC3-I, II, beclin-1, and *p*62, and (**B**) the LC3-II/I, beclin-1/β-actin, and *p*62/β-actin ratios were calculated. The experiment was performed independently in triplicate. * *p* < 0.05 vs the control.

**Figure 4 molecules-24-03230-f004:**
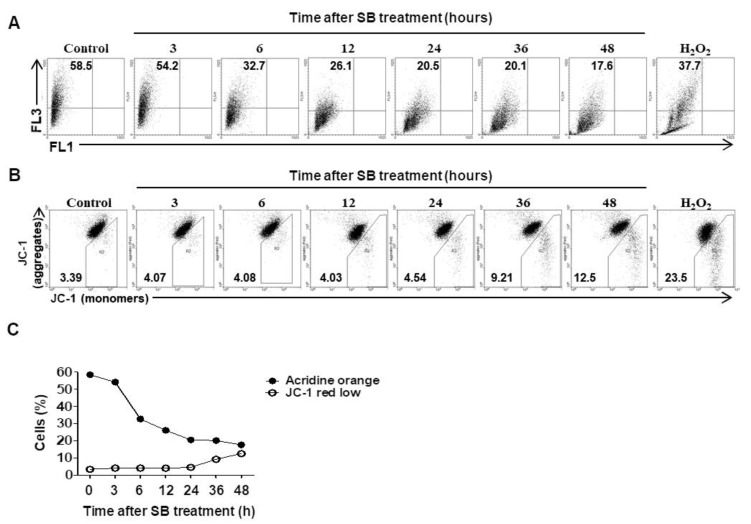
SB365 deteriorated lysosomal stability and mitochondrial membrane potential (MMP) in U87-MG cells. (**A**) SB365 induced lysosomal pH neutralization in U87-MG cells. Cells were treated with 10 μM SB365 for the indicated times, stained with (**A**) 3 μg/mL acridine orange for lysosomal stability measurement. (**B**) SB365 induced mitochondrial depolarization in U87-MG cells. Cells were stained with 2.5 μM JC-1 for 20 min for MMP measurement, harvested, and analyzed by flow cytometry. Cells treated with 0.5 mM H_2_O_2_ for 2 h constituted the positive control. (**C**) Combination of (**A**) and (**B**). The experiment was performed independently in triplicate.

**Figure 5 molecules-24-03230-f005:**
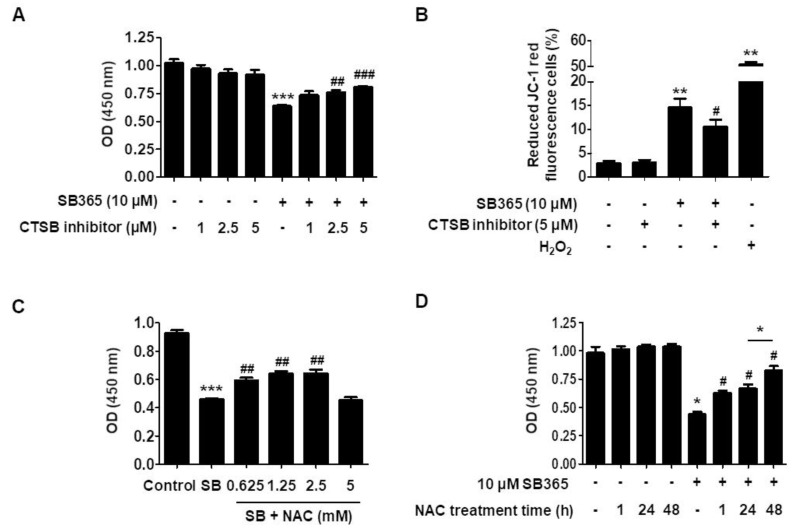
SB365 induced cell death via cathepsin B and ROS in U87-MG cells. (**A**) A cathepsin B inhibitor partially restored inhibited proliferation of U87-MG cells induced by SB365. Cells were cultured in 96-well plates, treated with 10 μM SB365 for 72 h in the presence of the indicated concentrations of cathepsin B inhibitor, and subjected to CCK-8 assay. (**B**) A cathepsin B inhibitor partially recovered SB365 induced MMP deterioration. U87-MG cells were treated with 10 μM SB365 for 72 h in the presence of 5 μM cathepsin B inhibitor, stained with JC-1, and MMP was analyzed by FACS. (**C**) NAC partially reduced the anti-proliferative effect of SB365 in U87-MG cells. Cells were cultured in 96-well plates, treated with 10 μM SB365 for 72 h in the presence of the indicated concentrations of NAC, and subjected to CCK-8 assay. NAC was added to the culture medium 1 h after SB365 treatment. (**D)** The same experiments were performed as in (**C**) with 2.5 mM NAC. However, NAC was treated 24 and 48 h after SB365 treatment, in addition to 1 h treatment. Quadruplicate samples were analyzed independently in triplicate. * *p* < 0.05, ** *p* < 0.01, *** *p* < 0.001 vs the control; # *p* < 0.05, ## *p* < 0.01, and ### *p* < 0.001 vs the SB365 group. CTSB, cathepsin B; NAC, *N*-acetyl cysteine.

**Figure 6 molecules-24-03230-f006:**
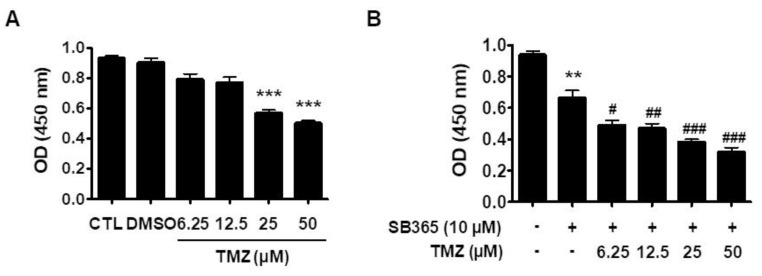
SB365 augmented the cytotoxic effect of TMZ on U87-MG cells. Cells were cultured in 96-well plates, treated with the indicated concentrations of TMZ in the (**A**) absence or (**B**) presence of 10 μM SB365, and subjected to CCK-8 assay. Quadruplicate samples were analyzed independently in triplicate. ** *p* < 0.01, and *** *p* < 0.001 vs the control; # *p* < 0.05, ## *p* < 0.01, and ### *p* < 0.001 vs the SB365 group. CTL, control; TMZ, temozolomide.

**Figure 7 molecules-24-03230-f007:**
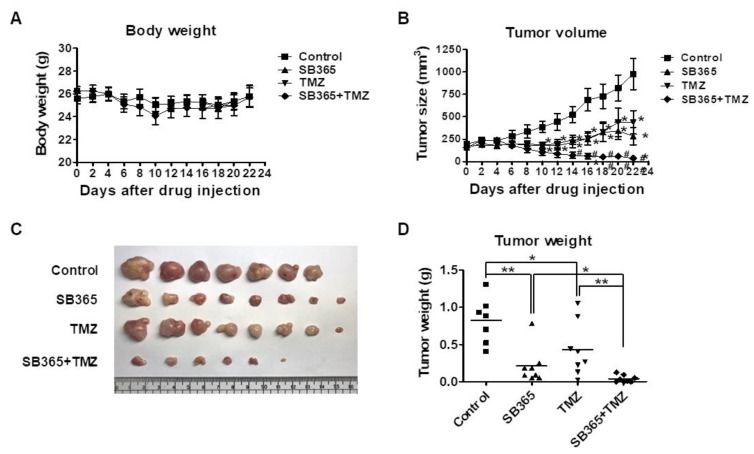
Combination of SB365 with TMZ additively suppressed the growth of U87-MG tumors in a mouse xenograft model. U87-MG cells were subcutaneously inoculated into both flanks of nude mice. When the tumor reached a volume of ~100–200 mm^3^, mice were intratumorally administered with SB365 every other day and/or with TMZ intraperitoneally daily for 22 days. The control received vehicle (<3% DMSO). (**A**) Body weight and (**B**) tumor size were measured every other day. The mice were euthanized, and (**C**) the tumors were extracted and (**D**) weighed. *n* = 8 per group. * *p* < 0.05 and ** *p* < 0.01 vs the control. SB365, SB365-treated group; TMZ, temozolomide-treated group.

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
