# Peer review of "SB365, Pulsatilla Saponin D Induces Caspase-Independent Cell Death and Augments the Anticancer Effect of Temozolomide in Glioblastoma Multiforme Cells"

_molecules, 2019, doi:10.3390/molecules24183230_

Round 1
Reviewer 1 Report
This paper reports some interesting experimental results, but it is not easy to read partly, but not only, due to several grammatical and typo errors. I have some concerns about all the manuscript sections, as described in the following.
Introduction. The saponin D SB365 is already known for its antitumor activity in in vitro and in vivo systems. In particular, SB365 is reported to induce in vitro cell death by apoptosis in some tumor cell lines (unspecified), and to inhibit authophagy in others (in this case cell lines are specified, but please notice that MCF7 are breast cancer cells and not liver cancer cells). Being this paper largely based upon in vitro experiments on tumor cell lines (GBM), I would strongly recommend to give more details about published data (tumor cell lines, human/murine, histotype, biological/pharmacological end-point) in the introduction section (rather than in the discussion) so that the reader may clearly understand the background, as well as the aim of the work. The choice of GBM cell lines used in this work must be justified. More attention should be paid to the description of SB365 pharmacological actions: (i) “SB365 is reported…to inhibit the autophagic flux in HeLa…[19](3rd paragraph)”; (ii) “SB365 induces autophagy in HeLa cells…[19] (discussion, 5th paragraph).
RESULTS. 2.1.SB365 and GBM cell proliferation. It would be better to calculate and report IC50 values. As far as the annexin V/7-AAD assay, it is quite clear that a 24-h treatment has no effect, whereas a dose-dependency can be observed after 48 and 72 h treatment. However, a strict adherence to the measured parameters would be preferable in results description: e.g. annexin V-positive cells instead of “damaged” cells (figure 1 caption), “dead” cells (text), “apoptotic cells” (paragraph 2.2).
2.2. caspase-independent cell death. The first sentence is ambigous: does “early apoptotic cells” refer to 24-h treatment? What is the context of ref. [20]? Cell growth inhibitory potency of SB365 (IC50) on the three cell lines should be provided. It is necessary to describe the morphological featurs of nuclei (fragmented, normal…), giving more details in the absense of quantitative analysis (fig. 2c).
2.3. Autophagy. The reason why the measured parameters are good indicators of autophagic flux must be explained. It is not clear if LC3-I/II and p62 expression was evaluated by WB at distinct time points during a 24-h treatment (fig. 3a): in that case the text should be corrected (page 5, 2nd sentence; fig. 3 caption).
2.4. The first sentence is rather inconclusive. The methodological bases behind the various probes must be summarized, and results described.
2.6. An appropriate drug combination study (experimental design, data acquisition, data interpretation, and computerized simulation) should be performed to demonstrate authors’ claim.
Discussion. It would be better to shorten this section: some parts in the introduction and others in the results section.
Figure legends. I’m not totally convinced by the unconventional style adopted by the authors. In any case all figure legends should be more detailed (in particular flow cytometry and microscopy).
Author Response
Dear;
We did all that we could do to meet the reviewers’ comments.
Besides,
We added a figure of a result [Fig. 5 D], which we got after we had submitted the manuscript, in association with the comment #18. The new figure file is uploaded.
We changed the Acknowledgements:
→ This research was supported by the Education and Research Encouragement Fund of Seoul National University Hospital (2019).
The English in this document has been checked by at least two professional editors, both native speakers of English before it was submitted. For a certificate, please see: http://www.textcheck.com/certificate/frGtD9
Reviewer 1.
Introduction. The saponin D SB365 is already known for its antitumor activity in in vitro and in vivo systems. In particular, SB365 is reported to induce in vitro cell death by apoptosis in some tumor cell lines (unspecified), and to inhibit autophagy in others (in this case cell lines are specified,
1) please notice that MCF7 are breast cancer cells and not liver cancer cells).
→ Thank you for pointing out our mistake. We corrected it.
2) Being this paper largely based upon in vitro experiments on tumor cell lines (GBM), I would strongly recommend to give more details about published data (tumor cell lines, human/murine, histotype, biological/pharmacological end-point) in the introduction section (rather than in the discussion) so that the reader may clearly understand the background, as well as the aim of the work.
3) The choice of GBM cell lines used in this work must be justified.
→ These two issues seem to be intimately related: We added the sentences at the end of the Introduction.
“To this end, we selected two GBM cell lines, U87-MG and T98G, among dozens of them. These are of human type 4 glioma cells [20]. We selected them because they are the most extensively employed ones in related studies [21], and especially they possess opposite characteristics concerning the susceptibility to TMZ. U87-MG cells are susceptible to TMZ, while T98G cells are not. T98G cells express O6-methylguanine-DNA methyltransferase (MGMT), which removes the methyl group at the O6 position of guanine added by TMZ [22], rendering them resistant to this drug. The survival duration of patients with MGMT-expressing GBM is approximately 2 years less than that of patients with non-functional methylated MGMT genes [23].”
4) More attention should be paid to the description of SB365 pharmacological actions: (i) “SB365 is reported…to inhibit the autophagic flux in HeLa…[19](3rd paragraph)”; (ii) “SB365 induces autophagy in HeLa cells…[19] (discussion, 5th paragraph).
→ It is somewhat tricky. Actually, the article [19] reported that SB365 was “an inducer of autophagosome formation, but an inhibitor of autophagic flux.” Thus, to avoid misunderstanding, we added “though it inhibits subsequent autophagic flux” at the end of the sentence in the Discussion “SB365 induces autophagy in HeLa cells by increasing ERK phosphorylation and decreasing mTOR activation.”
5) RESULTS. 2.1.SB365 and GBM cell proliferation. It would be better to calculate and report IC50 values.
→ We added the value in the Result 2.1., and also added the method in the Materials and Methods 4.3. as below;
“Calculated half maximal inhibitory concentration (IC50) for 72 h treatment was 8.9 μM.”
“IC50 value was obtained, based on the CCK-8 results, by the Quest Graph™ IC50 Calculator, a four parameter logistic regression model [52], with the minimum response value was fixed to zero computationally.”
6) As far as the annexin V/7-AAD assay, it is quite clear that a 24-h treatment has no effect, whereas a dose-dependency can be observed after 48 and 72 h treatment. However, a strict adherence to the measured parameters would be preferable in results description: e.g. annexin V-positive cells instead of “damaged” cells (figure 1 caption), “dead” cells (text), “apoptotic cells” (paragraph 2.2).
→ We changed all the expressions such as “damaged cell”, “dead cells”, and “apoptotic cells to “annexin V-positive cells”
7) 2.2. caspase-independent cell death. The first sentence is ambigous: does “early apoptotic cells” refer to 24-h treatment? What is the context of ref. [20]?
→ We meant by “early apoptotic cells” those cells in the early stage of the apoptotic process, which are 7-AAD (−) Annexin-V (+). To make it clear, we changed “7-AAD (−) Annexin V (+) early apoptotic cells” with “cells in the early stage of the apoptotic process, which are 7-AAD-negative and Annexin V-positive [24]”
We also replaced the reference 20 with
[Vermes I, Haanen C, Reutelingsperger C. Flow cytometry of apoptotic cell death. J Immunol Methods. 2000 Sep 21;243(1-2):167-90.]
8) Cell growth inhibitory potency of SB365 (IC50) on the three cell lines should be provided.
→ We calculated IC50 of SB365 for the three cell lines
. and the numbers are presented in the Figure Legend 2 as follows;
U87-MG, HT-29 (1 × 105/well), and Huh-7 cells (1 × 105/well) in six-well plates were treated with 10, 5, and 15 μM SB365, respectively, which exerted cytotoxic effects of similar magnitudes on the respective cell lines. → “U87-MG, HT-29 (1 × 105/well), and Huh-7 cells (1 × 105/well) in six-well plates were treated with 10, 5, and 15 μM SB365, respectively. The calculated IC50 values of SB365 on each cell line were 8.9, 5.1, and 13.2 μM, respectively.”
9) It is necessary to describe the morphological features of nuclei (fragmented, normal…), giving more details in the absence of quantitative analysis (fig. 2c).
→ We changed the previous sentence “DAPI staining showed nuclear condensation in HT-29 and Huh-7 cells, but not in U87-MG cells (Figure 2C)” to that below with some quantitative concept. We counted the number of fragmented and/or blebbed nuclei in SB-365 treated HT-29 and Huh-7 cells, but the n number, i.e. the number of the microscopic fields that were counted, is not enough to mention the exact % of the cells with morphologically deformed nuclei. We did not feel the necessity to present the “%”, because belbbed or fragmented nuclei were not observed in the SB365-treated U87-MG cells, which, we thought, was the point of this experiment.
“When the cells were stained with DAPI, SB365-treated HT-29 and Huh-7 cells showed nuclear blebbing and/or fragmentation with a frequency of 1~4 nuclei per a high-power field. However, SB365-treated U87-MG cells showed round or oval nuclei without blebbing and fragmentation.”
10) 2.3. Autophagy. The reason why the measured parameters are good indicators of autophagic flux must be explained.
→ We added the background of the method in 2.3. as below;
When autophagy is induced, microtubule-associated protein light chain 3 (LC3)-I is converted to LC3-II in combination with phosphatidylethanolamine in the cytosol to produce autophagosomes, and p62 binds to ubiquitinated proteins and pulls them into autophagosomes to be decomposed due to subsequent autophagic flux [27]. When the autophagic flux is inhibited, LC3-II and p62 accumulate in the cell [28]. Thus, LC3-II and p62 have been regarded as indicators of autophagic flux inhibition.
11) It is not clear if LC3-I/II and p62 expression was evaluated by WB at distinct time points during a 24-h treatment (fig. 3a): in that case the text should be corrected (page 5, 2ndsentence; fig. 3 caption).
→ It’s a mistake. We appreciate the reviewer for the meticulous check. We corrected it as “within 24 h.”
12) 2.4. The first sentence is rather inconclusive.
The methodological bases behind the various probes must be summarized, and results described.
→ The sentence was corrected as below:
“Because inhibition of autophagic flux is associated with lysosomal dysfunction such as neutralization and permeabilization [29], we performed a lysosomal stability test. Cells were stained with acridine orange and analyzed by flow cytometry.”
→ Explanation for the methodological bases is added in the Materials and Methods.
4.6. Lysosome stability assay
Lysosomal membrane stability was determined by staining SB365-treated cells with 3 μg/mL acridine orange (A8097; Sigma) for 20 min at 37°C, followed by determination of the red (FL3; 650 nm) and green (FL1; 510–530 nm) fluorescence of cells excited by blue (488 nm) light using a FACSCalibur instrument.
→ “Lysosomal membrane stability was determined by staining SB365-treated cells with 3 μg/mL acridine orange (A8097; Sigma) for 20 min at 37°C. This metachromatic dye emits red fluorescence when it is confined in the cytosol where it is present as a monomer. When the dye penetrates into the dysfunctional lysosome, it converts into aggregates due to the acidic environment in the lysosome and emits green fluorescence. The property has been used to measure lysosomal membrane stability [53]. Flow cytometric analysis was performed to determine the red (FL3; 650 nm) and green (FL1; 510–530 nm) fluorescence of cells excited by blue (488 nm) light using a FACSCalibur instrument.”
4.7. Mitochondrial membrane potential assay
SB365-treated cells were stained with 2.5 μM JC-1 [56] (T3168; Life Technologies, Carlsbad, CA, USA) for 20 min at 37°C, and analyzed by flow cytometry. FL-1 and FL-2 fluorescence was monitored from monomeric JC-1 and JC-1 aggregates, respectively. Red and green fluorescent JC-1 aggregates indicate higher and lower mitochondrial membrane potential (MMP), respectively [56].
→ “SB365-treated cells were stained with 2.5 μM JC-1 (T3168; Life Technologies, Carlsbad, CA, USA) for 20 min at 37°C, and analyzed by flow cytometry. JC-1 is a lipophilic and cationic dye. It enters the mitochondria, converts from monomers to aggregates by membrane potential, and accumulates inside the mitochondrion. In FACS analysis, monomers and aggregates emit green and red fluorescence, and indicate lower and higher mitochondrial membrane potential (MMP), respectively [54].”
13) 2.6. An appropriate drug combination study (experimental design, data acquisition, data interpretation, and computerized simulation) should be performed to demonstrate authors’ claim.
→ Please understand that we have no time to perform that kind of sophisticated experiment for now. That should be our future study. Considering this limitation, we concluded that SB365 COULD be used in combination with TMZ, not CAN be used.
14) Discussion. It would be better to shorten this section: some parts in the introduction and others in the results section.
→ We tried to shorten the Discussion with a limited outcome. Additionally we re-write much of the Discussion to make it easier to read.
We deleted the sentences below;
“We evaluated the cytotoxic effects of SB365 on U87-MG and T98G human GBM cells. As expected”
“When LMP occurs, proteases such as cathepsins B and D are released from the lysosome, leading to Bax-dependent [38] or -independent40 mitochondrial damage. Treatment with SB365 resulted in significant deterioration of MMP after 36 h (Fig. 4b, c), which was partially recovered by a cathepsin B inhibitor (Fig. 5b). There was a delay between changes in LMP and MMP (6 h), possibly due to delayed release [39] or activation [40] of cathepsins.”
“Thus, suppression of the NF‑κB/MGMT pathway by drug combinations is used to overcome TMZ resistance [54].”
The sentence below was shortened;
“Following deterioration of MMP, cytochrome c46, AIF47, EndoG48, and HtrA2/Omi49 are released from the damaged outer mitochondrial membrane. Cytochrome c activates caspase‑3/9, leading to apoptosis, endonuclease G enters the nucleus and degrades DNA, and AIF induces chromatin condensation (reviewed in Tait et al., 201450). However, in this study an antioxidant markedly restored the SB365‑induced reduction in cell proliferation (Figs. 5c, S6), implying that deterioration of MMP contributed to SB365‑induced death in GBM cells in a manner involving excess ROS.”
→ “Indeed, the phenomena caused by various factors secreted from the mitochondria when MMP deterioration occurs, such as the activation of caspase‑3/9 leading to apoptosis by cytochrome c, degrading DNA by endonuclease G, and chromatin condensation by AIF [47] were not observed in this experiment. Another substance that is released from deteriorated mitochondria is ROS.”
We moved the sentence from the Discussion to the Introduction with some revision;
“T98G cells express O6-methylguanine-DNA methyltransferase (MGMT), which removes the methyl group at the O6 position of guanine added by TMZ [50], thus rendering T98G cells resistant to TMZ [51]. MGMT expression is required for TMZ resistance [52], and the survival duration of patients with MGMT-expressing GBM is approximately 2 years less than that of patients with non-functional methylated MGMT genes [53].“ → “T98G cells express O6-methylguanine-DNA methyltransferase (MGMT), which removes the methyl group at the O6 position of guanine added by TMZ [22], rendering them resistant to this drug. The survival duration of patients with MGMT-expressing GBM is approximately 2 years less than that of patients with non-functional methylated MGMT genes [23].”
15) Figure legends. I’m not totally convinced by the unconventional style adopted by the authors. In any case all figure legends should be more detailed (in particular flow cytometry and microscopy).
→ We re-described some of the Figure legend in more detail.
Figure 1. SB365 exerted a cytotoxic effect on U87-MG cells. (A-C) SB365 inhibited the proliferation of U87-MG cells. The cells in 96-well plates were treated with SB365 at the indicated concentrations for (A) 24, (B) 48, or (C) 72 h in quadruplicate, and subjected to CCK‑8 assay. (D, E) SB365 increased the frequency of the annexin-V positive cells. U87-MG cells in six-well plates were treated as above, stained with Annexin V and 7‑AAD, and subjected to FACS analysis. (D) A representative FACS profile after 72 h and (E) the frequency of annexin-V positive cells. Experiments were performed independently in triplicate. * p < 0.05, ** p < 0.01, and *** p < 0.001 vs. the control.
Figure 2. SB365 induced caspase-independent death in U87-MG cells. U87-MG, HT-29 (1 × 105/well), and Huh-7 cells (1 × 105/well) in six-well plates were treated with 10, 5, and 15 μM SB365, respectively. The calculated IC50 values of SB365 on each cell line were 8.9, 5.1, and 13.2 μM, respectively. (A) Cell lysates were subjected to western blotting of caspase‑3 cleavage, (B) followed by densitometry. (C) SB365 induced nuclear fragmentation in HT-29 and Huh-7 cells, but not in U87-MG cells. Cells were treated with 10 μM SB365 for 72 h, adhered to an eight-well multispot slide, and stained with DAPI (blue). Arrows indicate fragmented nuclei. Images were acquired using a fluorescence microscope. The scale bar represents 50 μm. CTL, control group; SB, SB365-treated group.
Figure 4. SB365 deteriorated lysosomal stability and mitochondrial membrane potential in U87-MG cells. (A) SB365 induced lysosomal pH neutralization in U87-MG cells. Cells were treated with 10 μM SB365 for the indicated times, stained with 3 μg/mL acridine orange for lysosomal stability measurement. (B) SB365 induced mitochondrial depolarization in U87-MG cells. Cells were stained with 2.5 μM JC-1 for 20 min for MMP measurement, harvested, and analyzed by flow cytometry. Cells treated with 0.5 mM H2O2 for 2 h constituted the positive control. (C) Combination of (A) and (B). The experiment was performed independently in triplicate.
Figure 5. SB365 induced cell death via cathepsin B and ROS in U87-MG cells. (A) A cathepsin B inhibitor partially restored inhibited proliferation of U87-MG cells induced by SB365. Cells were cultured in 96-well plates, treated with 10 μM SB365 for 72 h in the presence of the indicated concentrations of cathepsin B inhibitor and subjected to CCK-8 assay. (B) Cathepsin B inhibitor partially recovered SB365 induced MMP deterioration. Cells were treated with 10 μM SB365 for 72 h in the presence of 5 μM cathepsin B inhibitor and MMP was analyzed by FACS. (C) NAC partially reduced the anti-proliferative effect of SB365 in U87-MG cells. Cells were treated with 10 μM SB365 for 72 h in the presence of the indicated concentrations of NAC, and subjected to a CCK-8 assay. NAC was added to the culture medium 1 h after SB365 treatment. (D) The same experiments were performed as in (C) with 2.5 mM NAC. However, NAC was treated 24 and 48 h after SB365 treatment, in addition to 1 h treatment. Quadruplicate samples were analyzed independently in triplicate. ** p < 0.01, *** p < 0.001 vs. the control; # p < 0.05, ## p < 0.01, and ### p < 0.001 vs. the SB365 group. CTSB, cathepsin B; NAC, N‑acetyl cysteine.

Reviewer 2 Report
Molecules (molecules-580251), Comments to the Authors:
Title: SB365, Pulsatilla saponin D induces caspase-independent cell death and augments the anticancer effect of temozolomide in glioblastoma multiforme cells
Comments
The submitted manuscript discussed the effects of SB365 on U87-MG and T98G glioblastoma multiforme (GBM) cells, and its efficacy in combination with temozolomide for treating GBM. SB365 exerted a cytotoxic effect on GBM by triggering caspase-independent cell death. Inhibition of autophagic flux and neutralization of the lysosomal pH occurred rapidly after application of SB365, followed by deterioration of mitochondrial membrane potential. A cathepsin B inhibitor and N-acetyl cysteine, an antioxidant, partially recovered cell death induced by SB365. SB365 in combination with temozolomide exerted a synergistic cytotoxic effect in vitro and in vivo.
I think the manuscript can be accepted for publication after the authors respond to the following comments:
More than seven manuscripts discussed the cytotoxic effect of SB365 on different cell lines. In each manuscript, the authors reached a different conclusion on the cytotoxic mechanism of action. In the current manuscript, the authors found that the cytotoxic effect is not related to apoptosis which was not the case with previous manuscripts on other cell lines. It seems that SB365 has many approaches to kill cytotoxic cells, which renders it difficult to be further developed into a clinical drug. Also, saponins with their sugar part are difficult to be developed into clinical drugs due to their side effects. Can the authors comment on these facts? Why did not the authors use positive control to compare the inhibition of autophagy to the SB365? The authors found that SB365 caused MMP disturbance by 10% compared with the control. This 10% is a low value to consider the cytotoxic effect of SB365 is partially through the disturbance in MMP. Can the authors comment on this low value? The authors should discuss why NAC did not recover cellular proliferation at 5 micromoles? The following sentences are not clear and need rephrasing, “Similarly, inhibition of cathepsin B but not cathepsin D, restored the paclitaxel-, epothilone B-, and discodermolide-induced death of human non-small cell lung cancer cells [41] and supraoptimally activated T cells [42]. Although based on their similar molecular weights cathepsin B and D may have been released simultaneously the release of only cathepsin B cannot be ruled out [43] These results suggest These results suggest that the roles of cathepsins vary depending on cell context [24]. However, further study is required.”Author Response
Dear;
We did all that we could do to meet the reviewers’ comments.
Besides,
We added a figure of a result [Fig. 5 D], which we got after we had submitted the manuscript, in association with the comment #18. The new figure file is uploaded.
We changed the Acknowledgements:
→ This research was supported by the Education and Research Encouragement Fund of Seoul National University Hospital (2019).
The English in this document has been checked by at least two professional editors, both native speakers of English before it was submitted. For a certificate, please see: http://www.textcheck.com/certificate/frGtD9
Reviewer 2.
1) More than seven manuscripts discussed the cytotoxic effect of SB365 on different cell lines. In each manuscript, the authors reached a different conclusion on the cytotoxic mechanism of action. In the current manuscript, the authors found that the cytotoxic effect is not related to apoptosis which was not the case with previous manuscripts on other cell lines.
It seems that SB365 has many approaches to kill cytotoxic cells, which renders it difficult to be further developed into a clinical drug.
Also, saponins with their sugar part are difficult to be developed into clinical drugs due to their side effects. Can the authors comment on these facts?
→ The fact that the mechanisms of cancer cell death triggered by SB365 differ in different cancer cells can be an obstacle in developing a widely applicable global anticancer drug for many different kinds of cancers. However, it would not, if the drug targets specific cancer, for example, such as glioblastoma.
Meanwhile, as the reviewer pointed out, the side effects caused by the sugar domain should be considered in future drug development. Even though the clinical use of SB365 in 50 pancreatic cancer patients does not report any hematologic side effects [Moon et al., 2015], the risk should be further monitored. We added the following sentence at the end of the Discussion.
“One concern is that SB365 exerts hemolytic toxicity on red blood cells of the sheep [42] and the rabbit [51], which was considered as a major drawback for its clinical development [42].
2) Why did not the authors use positive control to compare the inhibition of autophagy to the SB365?
→ Actually, we had performed western blotting with U87-MG cell lysates treated with bafilomycin A1 (the figure below), which is known to be an inhibitor of autophagy flux [Int J Cancer. 2009;124(5):1060-71.], and got positive results. We thought that the reliability of the experimental method had been secured, and that there were a lot of Figures in the paper, so we did not put this Figure in the manuscript.
18) The authors found that SB365 caused MMP disturbance by 10% compared with the control. This 10% is a low value to consider the cytotoxic effect of SB365 is partially through the disturbance in MMP. Can the authors comment on this low value?
→ We agree that 10% is not enough value to explain the cell death by SB365 by itself. When MMP occurs, factors such as cytochrome c, AIF, and EndoG are released and cytochrome c activates caspase‑3/9, leading to apoptosis, endonuclease G enters the nucleus and degrades DNA, and AIF induces chromatin condensation [48], all of which were not observed in our experiment. Thus, we came to think of ROS, which is known to be leaked out from the deteriorated mitochondria [30]. Substantial to this assumption, 2.5 mM NAC recovered the cytotoxicity by 50% when added 1 h after SB365 treatment. Additionally, we concerned two points. The first one was that the MMP deterioration started at 24 h and gradually increased thereafter up to 48 h. The second point was that there was no information about how much NAC persisted in culture media without decomposition. Thus, we treated NAC 24 and 48 h after SB365 treatment instead of I h [Fig. 5D, newly added]. Results showed that nearly 70% of cell death was recovered when NAC was added after 48 h. These results imply that NAC decomposes a little (even though we could not determine the exact amount) in culture media, and at the same time, that the main factor leading to cell death by SB365 is ROS which presumably begin to accumulate after 24 h in parallel with MMP deterioration.
Accordingly, we added and revised the Results and the Discussion as flows. Also, we changed the Conclusion in a more specific way;
In the Result, “Considering that MMP deterioration started late during the experiment time (Figure 4B and C) and that NAC could decompose in culture media, we performed the same experiment with 2.5 mM NAC, which at this time was added 24 and 48 h after SB365 treatment, instead of 1 h after (Figure 5D). As a result, NAC recovered the cytotoxicity by SB365 up to over 70% when added at 48 h.”
In the Figure Legend 5, “(D) The same experiments were performed as in (C) with 2.5 mM NAC. However, NAC was treated 24 and 48 h after SB365 treatment, in addition to 1 h treatment.”
In the Discussion, “The frequency of the cells with MMP deterioration was only 5.8% at 36 h and 8.6% at 48 h after SB365 treatment (Figure 4 B and C). These are low values to consider that MMP deterioration directly led to the SB365-induced cell death. Indeed, the phenomena caused by various factors secreted from the mitochondria when MMP deterioration occurs, such as the activation of caspase‑3/9 leading to apoptosis by cytochrome c, degrading DNA by endonuclease G, and chromatin condensation by AIF [47] were not observed in this experiment. Another substance that is released from deteriorated mitochondria is ROS. Autophagic flux inhibition, which was induced by SB365 in GBM cells in this experiment, leads to the accumulation of ROS [34,35]. Excess ROS accelerate lysosomal permeabilization, and leaked lysosomal proteases deteriorate MMP, resulting in increased cytoplasmic ROS leakage, creating a vicious cycle [48]. Thus, ROS could be a factor for the SB365-induced cytotoxicity. Substantial to this assumption, 2.5 mM NAC recovered the cytotoxicity by over 50% when added 1 h after SB365 treatment (Figure 5C). Furthermore, when NAC was added 48 h after SB365 treatment, the recovery rate was over 70% (Figure 5D). These results imply that ROS was the main factor leading to cell death by SB365, and ROS presumably began to accumulate to cause cell death 24 h after SB365 treatment in parallel with MMP deterioration.”
In the Conclusion; “SB365 inhibited autophagic flux, and induced CICD in GBM cells in a manner mediated by cathepsin B and mainly by ROS very likely due to autophagic flux inhibition and MMP deterioration.”
In the Abstract; “In conclusion, SB365 inhibits autophagic flux and induces caspase‑independent cell death in GBM cells in a manner involving cathepsin B and mainly reactive oxygen species, and its use in combination with temozolomide shows promise for the treatment of GBM.
Added Figure (5 D);
19) The authors should discuss why NAC did not recover cellular proliferation at 5 micromoles?
→ ROS is not just a harmful byproduct of metabolism but functions as a signal messenger such as in cell proliferation [Free Radic Biol Med. 2016, 100:86-93]. Thus, excessive eradication of ROS could result to malfunction of the cells. We added some to the Discussion as below.
“Meanwhile, 5 mM NAC failed to recover cell proliferation. This may be because of excessive ROS eradication, which performs physiological functions in cell proliferation [49]. Substantial to this assumption, 10 mM NAC augmented the cytotoxic effect of SB365 (data not shown). Also, even the low concentrations of NAC (0.623 ~ 2.5 mM) augmented cell proliferation inhibition by SB365 when it was added to the culture media before SB365 (data not shown).”
20) The following sentences are not clear and need rephrasing, “Similarly, inhibition of cathepsin B but not cathepsin D, restored the paclitaxel-, epothilone B-, and discodermolide-induced death of human non-small cell lung cancer cells [41] and supraoptimally activated T cells [42]. Although based on their similar molecular weights cathepsin B and D may have been released simultaneously the release of only cathepsin B cannot be ruled out [43]. These results suggest that the roles of cathepsins vary depending on cell context [24]. However, further study is required.”
→ We re-wrote the above sentences as follows; “In our results, a cathepsin B inhibitor restored cell death but a cathepsin D inhibitor did not (data not shown). Given that the molecular weights of cathepsins B and D are similar [44], and thus the two molecules would have been released at the same time, the contradictive effect of each inhibitor would be somewhat unexpected. However, the same results have been reported in paclitaxel-, epothilone B-, and discodermolide-treated human non-small cell lung cancer cells [45] and supraoptimally activated T cells [46]. Possibly, only cathepsin B had been released [44]. Or, these results suggest the varying role of cathepsins depending on the cells [30]. The exact mechanism is remained to be solved.”

Reviewer 3 Report
The authors investigated the role of SB365 alone or in combination with TMZ on the antiproliferative effect of glioblastoma multiforme cells GBM in vitro and in vivo. In general the manuscript is well elaborated and correctly written.
Some detailed comments
Proliferative assays. Is OD450 proportional to the number of cells? The number of cells or % changes should be on the Y axis. In Discussion the authors state that “Activation of caspase-3 is a final step in both intrinsic and extrinsic pathways of caspase –dependent apoptosis”. This sentence is not a quite true. Caspase-3 is not the final execution factor. At the beginning of the discussion, the authors state that the combination shows an additive effect and at the end of the discussion and in other parts of the work they state that SB365 and TMZ show a synergistic effect. The authors should set-up a Chou-Talalay (combination index) analysis to estimate if both compounds exert a “Synergistic” or an “Additive” effect.I recommend the manuscript to be published after MINOR REVISION.
Author Response
Dear;
We did all that we could do to meet the reviewers’ comments.
Besides,
We added a figure of a result [Fig. 5 D], which we got after we had submitted the manuscript, in association with the comment #18. The new figure file is uploaded.
We changed the Acknowledgements:
→ This research was supported by the Education and Research Encouragement Fund of Seoul National University Hospital (2019).
The English in this document has been checked by at least two professional editors, both native speakers of English before it was submitted. For a certificate, please see: http://www.textcheck.com/certificate/frGtD9
1) Proliferative assays. Is OD450 proportional to the number of cells? The number of cells or % changes should be on the Y axis.
→ As is well known, the Cell Counting Kit-8 (CCK-8) assay measures the intensity of yellow color in the culture media. The color is produced by the conversion of tetrazolium-8 monosodium salt (CCK-8) to its metabolite by living cells in the culture well. If we assume that all the cells in the culture well are in the same metabolic activity, the OD value at 450 nm surely reflects the cell number. However, we cannot guarantee the “assumption.” We only assume that the cells in the experimental groups are in a similar metabolic state and apply this method. Thus, we cannot exactly calculate the cell numbers from the OD values, but the OD values are believed to be proportional to the numbers of the cells.
So, we can put “% changes” on the Y-axis with the value of the control group being regarded as 100%, as the review recommended.
However, in that case, the concept of absolute value of the OD disappears. That is, for example, in Figure 1 of our results, the OD value of control groups in 24, 48, and 72 hours (Fig. 1 A, B, C) are increasing with culture days, which implies the cell proliferation and increased number of the cells with time. When we change the Y-axis from OD to % change, this feature disappears.
Thus, even though the reviewer's comment is appreciated, we think it's better to keep it as OD values.
2) In Discussion the authors state that “Activation of caspase-3 is a final step in both intrinsic and extrinsic pathways of caspase –dependent apoptosis”. This sentence is not a quite true. Caspase-3 is not the final execution factor.
→ What we really meant was that both the intrinsic and extrinsic pathways converge onto caspase-3, which implies that no activation of caspase-3 means no activation of both pathways. Thus, we re-wrote that sentence as follows; “Activation of caspase-3 is a converging step of both intrinsic and extrinsic pathways of caspase–dependent apoptosis [37].”
We also replaced the reference 21 with
[Khan KH, Blanco-Codesido M, Molife LR. Cancer therapeutics: Targeting the apoptotic pathway. Crit Rev Oncol Hematol. 2014 Jun;90(3):200-19.]
3) At the beginning of the discussion, the authors state that the combination shows an additive effect and at the end of the discussion and in other parts of the work they state that SB365 and TMZ show a synergistic effect. The authors should set-up a Chou-Talalay (combination index) analysis to estimate if both compounds exert a “Synergistic” or an “Additive” effect.
→ We performed a Chou-Talalay analysis and found that the drugs are synergistic in a confined interval. Thus we think that we cannot describe the drugs “synergistic.” Thus, we have to change all “synergistic” with “additive” for now. However, we have performed only one experiment concerning this issue due to the limited time. We will repeat more experiments. If we get appropriate data, we hope to reduce the change.

Round 2
Reviewer 1 Report
Overall, the Authors' answer is more than satisfactory.
Reviewer 2 Report
Molecules (molecules-580251, Revised Version), Comments to the Authors:
Title: SB365, Pulsatilla saponin D induces caspase-independent cell death and augments the anticancer effect of temozolomide in glioblastoma multiforme cells
Comments:
After reading the authors response to my comments and the revised manuscrtpt, I think the manuscript can be accepted for publication.